# Prevalence of birth injuries and associated factors among newborns delivered in public hospitals Addis Ababa, Ethiopia, 2021. Crossectional study

**Esubalew Amsalu Tibebu**[1]*, **Kalkidan Wondwossen Desta**[2], **Feven Mulugeta Ashagre**[2], **Asegedech Asmamaw Jemberu**[3]

**1** St.Peter Specialized Hospital, Addis Ababa, Ethiopia, **2** School of Nursing and Midwifery, College of Health Science, Addis Ababa University, Addis Ababa, Ethiopia, **3** Department of Medical Laboratory Sciences, College of Health Science, Addis Ababa University, Addis Ababa, Ethiopia

* esubemeklit2020@gmail.com

## Abstract

### Background

Birth injury is harm that a baby suffers during the entire birth process. It includes both birth asphyxia and birth trauma. In Ethiopia, birth injury has become the leading cause of neonatal morbidity and mortality, accounting around 28%-31.6% of neonatal mortality. The study aimed to assess the prevalence and factors associated with birth injuries among newborns delivered in public hospitals Addis Ababa, Ethiopia, 2021.

### Methods

Institution based cross-sectional study was conducted from February 15th to April 20th, 2021 in selected public hospitals of Addis Ababa, Ethiopia. Random sampling and systematic random sampling were used. Data was entered by using Epi data version 4.0.2 and exported in to SPSS Software version 25 for analysis. Both bivariate and multivariable logistic regressions analyses were used. Finally P-value <0.05 was used to claim statistically significant.

### Result

The prevalence of birth injury was 24.7%. In the final model, birth asphyxia was significantly associated with the short height of the mothers (AOR = 10.7, 95% CI: 3.59–32.4), intrapartal fetal distress (AOR = 4.74, 95% CI: 1.81–12.4), cord prolapse (AOR = 7.7. 95% CI: 1.45–34.0), tight nuchal cord (AOR = 9.2. 95% CI: 4.9–35.3), birth attended by residents (AOR = 0.19, 95% CI: 0.05–0.68), male sex (AOR = 3.84, 95% CI: 1.30–11.3) and low birth weight (AOR = 5.28, 95% CI: 1.58–17.6). Whereas, birth trauma was significantly associated with gestational diabetic mellitus (AOR = 5.01, 95% CI: 1.38–18.1), prolonged duration of labor (AOR = 3.74, 95% CI: 1.52–9.20), instrumental delivery (AOR = 10.6, 95% CI: 3.45–32.7) and night time birth (AOR = 4.82, 95% CI: 1.84–12.6).

**Data Availability Statement:** All relevant data are within the paper and its Supporting Information files.

**Funding:** The author(s) recived no specific funding for this work.

**Competing interests:** The authors have declared that no competing interests exist.

**Abbreviations:** ANC, Antenatal Care; APGAR, Appearance, Pulse rate, Grimace, Activity and Respiration; BMI, Body Mass Index; CI, Confident Interval; CPD, : Cephalopelvic disproportion; C/S, Caesarian Section; EDHS, Ethiopian Demographic Health Survey; GA, Gestational Age; GDM, Gestational Diabetes Mellitus; GMH, Gandhi Memorial Hospital; ICD, International classification of disease; OR, Odds Ratio; SPHMMC, St. Paul Hospital Millennium Medical College; TASH, Tikur Anbessa Specialized Hospital; WHO, World Health Organization; Y-12HMC, Yekatit 12 Hospital Medical College.

## Conclusion

The prevalence of birth injury among newborns has continued to increases and become life-threatening issue in the delivery and neonatal intensive care unit in the study area. Therefore, considering the prevailing factors, robust effort has to be made to optimize the quality obstetric care and follow up and emergency obstetrics team has to be strengthened to reduce the prevalence of birth injury.

## Introduction

The process of birth, whether spontaneous or assisted, is naturally traumatic for the newborns. Birth injury is the structural destruction or functional deterioration of the neonate's body due to a traumatic event at birth [1]. Birth related injuries encompass both those due to lack of oxygen (birth asphyxia) and physical trauma during the birth process (birth trauma). Both can occur separately or in combination [2–5].

Injuries to the newborns that result from mechanical forces (i.e. compression, traction) during the birth process are classified as mechanical birth trauma. Whereas, according to the World Health Organization (WHO), birth asphyxia defined as a "failure to initiate and sustain breathing at birth" [6] It's usually considered by low APGAR score: (Appearance, Pulse rate, Grimace, Activity and Respiration) <7 at 5th minutes, arterial cord pH < 7 and base deficit >12, neonate did not cry at birth or needed resuscitation, acidosis, seizure and hypotonia [7]. Study suggested that, birth asphyxia occur due to maternal antepartum, intra-partal and post partal factors [8]. Intra-partum related factors accounts the highest proportion of risk factors for birth asphyxia (70%). Whereas, antepartum and post partal factors accounts 20% and 10% respectively [9].

According to international classification of disease 10th revision (ICD-10) and different literature, the common types of birth injuries includes birth asphyxia and birth trauma (soft tissue injuries (bruises, petechial, subcutaneous fat necrosis, ulceration and perforation), extra cranial hemorrhages (cephalhaematoma, caput succedaneum, subgalial hemorrhage), intra-cranial hemorrhages, neurological injury (spinal cord injury, facial nerve palsy, brachial plexus injury such as Erb's palsy and Klumpke's palsy), musculoskeletal injury (long bone and clavicular fracture) [10–13].

According to 2016 WHO reports, it is estimated that 662, 000 neonatal deaths and 1.3 million stillbirths occur annually due to intra-partum related complications, or complications during labor and delivery. Birth injuries are among the three leading cause of most neonatal death worldwide which accounts for 10% of deaths in children under 5 years of age [14].

The incidence of birth injuries varies from place to place and it is mostly determined by the standard of available obstetrical management.

Birth asphyxia is a leading cause of brain damage and also survivors often experience life-long health problems like disabilities, developmental delays, palsy, intellectual disabilities and behavioral problems [15, 16].

In developed countries, the occurrences of birth injury are decreased due to the improvements in obstetric practice and care. In Ethiopia, according to 2019 mini EDHS (Ethiopia Demographic and Health Survey) reported, the percentages of delivery by skilled providers increased from 28% in 2016 to 50% in 2019. Despite of this, the number of neonatal death increased from 29 per 1000 live births to 30 per 1000 live births in Ethiopia [17].

Reports about the prevalence of birth injures among live birth newborns are limited in Ethiopia. As far as literature review revealed that, there is a limited research done on prevalence of

birth injuries among live birth delivery especially in the study area. However, intra-partum related complications among newborns during the time of delivery are still the leading cause of neonatal morbidity and mortality in Addis Ababa public hospitals. Therefore, this study was carried out to assess the prevalence of birth injuries and associated factors among newborns delivered in public hospitals Addis Ababa, Ethiopia.

## Material and methods

### Study design, study period and study area

Institutional based cross- sectional study was conducted in Addis Ababa Public Hospitals from February 15th to April 20th, 2021. This study was carried out in four randomly selected public hospitals (Tikur Anbessa Specialized Hospital (TASH), Yekatit 12 Hospital Medical College (Y-12HMC), Gandhi Memorial Hospital (GMH) and St. Paul Hospital Millennium Medical College (SPHMMC)).

### Study population

All live birth newborns delivered in selected public hospitals with gestational age of $\geq 28$ weeks were included in this study. Neonates with major congenital anomalies (like hydrops, congenital heart disease and neural tube defects), birth weight of <1000 g, those who have incomplete documentation (has no appropriate data that measure both maternal and early neonatal parameter) and mothers who are seriously ill and unable to respond to the question were excluded.

### Sample size and sampling procedure

The single population proportion formula was used to determine the sample size with the following assumptions: Where; **n** = Sample size, **Z** = 95% confidence level (Z $\alpha/2$ = 1.96), **α** = Level of significance 5% ($\alpha$ = 0.05) and **d** = Margin of error 5% (d = 0.05). The prevalence of birth trauma was (P) = 8.1% taken from the previous study done in Jimma University Specialized Hospital, South Western Ethiopia [18], and sample size was 125. The prevalence of birth asphyxia was (P) = 32.9% taken from the previous study conducted in Jimma zone public Hospitals, South West Ethiopia [19] after comparing with other studies done in Ethiopia [8, 20–22], After considering 10% non-response rate, the total sample size was **373.** Finally, from the calculated sample size for the first and second dependent variables, the largest sample size was **373.**

Simple random sampling technique was used to select four hospitals to be included in this study from 11 public hospitals. The number of study unit to be sampled from each selected hospital were determined by proportional to size allocation formula, based on three months report of delivery in each selected hospital. The study subject were selected from list of delivery registration book by using systematic random sampling technique every "K" value = 20, which was obtained through dividing the total number of delivery in three month report from selected hospital to the required sample size. Mothers that delivered more than one baby like twin, one of these babies was selected by using simple random sampling.

### Variables

The dependent variables of the study were birth injuries categorized as birth asphyxia and birth trauma. Whereas, the independent variables were socio demographic variables (maternal age in years, maternal weight, maternal height, pre-pregnancy body mass index (BMI), level of education, place of residence and marital status), medical and obstetrics variables (antenatal

care (ANC) follow up, pregnancy type, parity, chronic diabetic mellitus, gestational diabetes mellitus(GDM), chronic hypertension, pregnancy induced hypertension and abruption placenta), intrapartum variables (fetal presentation, duration of labor, cephalopelvic disproportion, intra-partal fetal distress, mode of delivery, cord prolapse, tight nuchal cord induction of labor, meconium stained amniotic fluids, premature rupture of membrane prolonged rupture of membrane, time of birth and qualification of birth attendant) and early neonatal variables (sex, birth weight, head circumferences, APGAR score, need of resuscitation and gestational age).

## Operational definitions

**Birth injury:** Injury to newborns that occur during labor and delivery who has diagnosis of birth trauma, birth asphyxia or both.

**Birth trauma:** Any physical injury to newborns during the entire birth process that can be recognized by clinical physical examination.

**Birth Asphyxia:** Failure to initiate, sustain breathing and not crying at birth and diagnosed based on Apgar score <7 at 5th minutes.

**Fetal distress:** When the fetal heart rate is either <100 or >180 beat/minutes or if there is non-reassurance fetal heart rate pattern.

**Major congenital anomalies:** Are structural or functional abnormalities which are significance effect to reduce life expectancy of newborns such as hydrops, congenital heart disease and neural tube defects.

**ANC follow up:** A programmed clinical visits of a mother at least one during her pregnancy in this study.

**Prolonged labor:** Defined as when the combined duration of the first and the second stage of labor are more than 12 hours in primipara or 8 hours in multipara mothers.

**Premature rupture of membrane:** Rupture of membrane of the amniotic sac and chorion occurred before onset of labor.

**Prolonged rupture of membrane:** Duration of rupture of membrane of the amniotic sac and chorion >18 hours till delivery.

## Data collection tools and procedures

Data collection tools were developed by reviewing different related literatures [8, 10, 18, 20, 22, 23]. Data was collected by Nurses and Midwives at delivery and post-natal ward by using structured interviewer administered questionnaire and checklist. The questionnaire was used to assess socio demographic characteristics of the mothers and medical and obstetrics variables of the mothers. The checklist was used to assess data on intra-partum and early neonatal variables. Birth injuries diagnosis obtained from mothers medical record which was diagnosed by gynecologist/obstetricians and residents. APGAR score was evaluated by resident and Gynacologist.

## Data processing and analysis

After completing data collection, data were categorized, coded, cleaned and recorded. The data was entered by using Epi data version 4.0.2 and exported in to SPSS software version 25. Descriptive statistical analysis such as frequencies, percentages, crosses tabulation and mean were performed. To assess the factors independently associated with birth injury, two regression models (considering the dependent variables to be (i) birth asphyxia and (ii) birth trauma) were used.

Bivariate logistic regression analysis was used to check the association between each independent variable with dependent variable. Then those variables with p-value ≤ 0.25 were entered a multivariable logistic regression model analysis in order to control the confounding factors. To check the correlation between independent variables, multi-colinearity (colinearity diagnostic taste) was done by using the value of variance inflation factors and tolerance. Hosmer and Lemeshow goodness of fit test and omnibus tests of model coefficients were done to test the fitness of the logistic regression in the final model, then it was found good (statistically insignificant value, $P$ value >0.05). The strength of association between dependent and independent variables was expressed by using adjusted odds ratio with 95% confidence interval. P-value <0.05 was considered as statistically significance. Finally, the findings were presented by using text, tables and graph.

### Ethical approval and informed consent

The research was reviewed and approved by School of Nursing and Midwifery, Addis Ababa University, College of Health Science, Institutional Review Board (IRB(Protocol number:52/21/ SNM)). Permission was also sought from each hospital. Study participants were asked for their willingness to participate in the study after explaining the purpose of the study. Then written informed consent was obtained from each participant. The privacy and confidentiality of information was strictly maintained by not writing the name of study participants on data collection tool.

## Results

### Socio demographic characteristics of the mothers

All of the 373 mothers were give an informed consent to participate with a response rate of 100%. The mean maternal age was 27.28 ± 5.16 SD years of whom 141 (37.8%) of mothers belonged to age groups of 25–29 years. The mean of BMI and height of the mothers were 22.65 ± 3.34SD kg/m$^2$ and 156.8 ± 8.5 SD cm respectively (Table 1).

### Medical and obstetric characteristics of the mothers

Among 373 study subjects, 367 (98.4%) of mothers attended ANC follow up during their pregnancy period. Majority of the participants, 312 (83.6%) had four and above ANC follow up. Half 186 (49.9%) of the mothers were primipara. Regarding the chronic medical illness of the mothers, majority of the participants 364 (97.6%) and 369 (98.9%) did not have chronic DM and hypertension respectively. Pregnancy induced hypertension 52 (14%) and gestational diabetes mellitus 40 (10.7%) were the most common obstetrics complication during pregnancy. Around one-tenth 39 (10.5%) of the participants who had pregnancy induced hypertension developed pre-eclampsia. Majority of the mothers 341 (91.4%) had single type of pregnancy and only 32 (8.6%) of the mothers had twin types of pregnancy (Table 2).

### Intrapartum related factors

According to the result of this study, majority 342 (91.7%) of the newborns were at vertex presentation. Around 88 (23.6%) of the newborns had intrapartum fetal distress. Among the total participated mothers, above two third 254 (68.1%) and 60 (16.1%) had spontaneous and induced onset of labor respectively. In addition to this, about 59 (15.8%) of the mothers did not experience any onset of labor during delivery i.e. delivered by elective cesarean section.

Nearly one third 119 (31.9%) of the mothers had prolonged duration of labor. Furthermore, one quarters 90 (24.1%), 54 (14.5%) and 81(21.7%) of the mothers faced premature rupture of membranes, prolonged rupture of membranes (≥18 hours) and meconium stained amniotic

**Table 1. Socio-demographic characteristics of mothers.**

| Variables | Category | Frequency (n) | Percentage (%) |
|---|---|---|---|
| **Age group of the mothers** | 15–19 | 22 | 5.9 |
| | 20–24 | 88 | 23.6 |
| | 25–29 | 141 | 37.8 |
| | 30–34 | 73 | 19.6 |
| | ≥35 | 49 | 13.1 |
| **Educational status** | No formal education | 51 | 13.7 |
| | Primary education | 133 | 35.7 |
| | Secondary education | 108 | 29.0 |
| | More than secondary | 81 | 21.6 |
| **Residency** | Urban | 358 | 96 |
| | Rural | 15 | 4 |
| **Marital status** | Married | 339 | 90.9 |
| | Divorced | 18 | 4.8 |
| | Single | 16 | 4.3 |
| **Height of the mother (in cm)** | <145 | 51 | 13.7 |
| | ≥145 | 322 | 86.3 |
| **BMI of the mothers (Kg/m$^2$)** | <18.5 (underweight) | 30 | 8 |
| | 18.5–24.9 (Normal) | 264 | 70.8 |
| | 25–29.9 (overweight) | 68 | 18.2 |
| | ≥30 (obese) | 11 | 2.9 |

**Key**: BMI: Body Mass Index

fluid respectively. More than half 217(58.2%) and 37(9.9%) of the newborns were delivered by cesarean section and instrumental delivery respectively. Regarding to cord problem, only 8 (2.1%) and 13 (3.5%) of the newborns developed cord prolapse and tight nuchal cord during delivery respectively. Majority of the delivery 184 (49.3%) and 135 (36.2%) attended by residents and midwifes respectively (Table 3).

## Early neonatal related factors

Of the total newborn babies, 225 (60.3%) of them were males. More than three quarters 288 (77.2%) of the newborn babies' gestational age was in the range of 37–42 weeks at birth. The mean gestational age at the time of birth was 39.45 ± 2.52 SD weeks. Besides, majority 285 (76.4%) of the participants had normal birth weight (2500–3999) gram and the average birth weight of the newborn babies was 3119.09 ± 649.25 SD grams. 336 (90.1%) of the participants had normal head circumference (33–38 cm) respectively. Moreover, around 52 (13.9%) of the newborns were unable to cry immediately after birth. About 321 (86.1%) of the newborn babies had normal Apgar score at fifth minutes after birth (7–10). Additionally, 43(11.5%) and 9 (2.4%) of the participants had moderate (4–6) and low (0–3) APGAR score respectively. Out of the study population,52 (13.9%) of the newborns needed resuscitation after birth (Table 4).

## Prevalence of birth injuries

The overall prevalence of birth injury was found to be 92 (24.7%) of the total study participants in this study. Birth asphyxia and birth trauma were identified in 52 (13.9%) and 48 (12.9%) of these babies, respectively. A total of eight newborns (2.1%) suffered from both birth asphyxia and birth trauma (Fig 1).

**Table 2. Medical and obstetrics characteristics of the mother.**

| Variables | Category | Frequency (n) | Percentage (%) |
|---|---|---|---|
| **ANC follow up** | Yes | 367 | 98.4 |
| | No | 6 | 1.6 |
| **Number of ANC follow up** | 1–3 | 55 | 14.7 |
| | ≥4 | 312 | 83.6 |
| **Facilities of ANC follow up** | Health centers | 262 | 70.2 |
| | Government hospitals | 78 | 20.9 |
| | Private hospitals | 19 | 5.1 |
| | Private clinic | 6 | 1.6 |
| | NGO clinic | 2 | 0.5 |
| **Parity** | Primipara | 186 | 49.9 |
| | Multipara | 187 | 50.1 |
| **Gravidity** | Primigravida | 160 | 42.9 |
| | Multigravida | 213 | 57.1 |
| **Types of pregnancy** | Single | 341 | 91.4 |
| | Twins | 32 | 8.6 |
| **Medical illness of the mothers** | | | |
| **Chronic DM** | Yes | 9 | 2.4 |
| | No | 364 | 97.6 |
| **Chronic hypertension** | Yes | 4 | 1.1 |
| | No | 369 | 98.9 |
| **HIV test done** | Yes | 373 | 100 |
| | No | 0 | 0 |
| **HIV Status** | Positive | 8 | 2.1 |
| | Negative | 365 | 97.9 |
| **Others*** | | 12 | 3.21 |
| **Obstetric complication of the mothers** | | | |
| **Gestational DM** | Yes | 40 | 10.7 |
| | No | 333 | 89.3 |
| **Pregnancy induced hypertension** | Yes | 52 | 14 |
| | No | 321 | 86 |
| **Types of pregnancy induced hypertension** | Pre-eclampsia | 39 | 10.5 |
| | Eclampsia | 13 | 3.5 |
| **Abruptio placenta** | Yes | 8 | 2.1 |
| | No | 365 | 97.9 |
| **Others**** | | 26 | 7 |

Key

* = Anemia, congestive heart failure, thrombocytopenia, asthma and hydronephrosis

** = Oligohydramnious and chorioamnionitis

ANC: Antenatal Care, DM: Diabetes Mellitus, HIV: Human Immunodeficiency virus, NGO: Non-governmental organization.

Among those newborns who diagnosed with birth trauma, the most common types were extra cranial trauma 39 (81.2%), neurological trauma 13 (27%) and soft tissue trauma 10 (21%). From extra cranial trauma, more than half, 20 (51.2%) and 10 (25.6%) of the newborns babies developed subgalial hemorrhage and cephalhaematoma respectively. Among neurological trauma and soft tissue trauma, the largest proportions contributed by facial palsy 8 (61.5%) and facial &skin bruises 5(50%) respectively. Furthermore, 14 (29.2%) newborns developed two types of birth trauma (Table 5).

**Table 3. Intra-partum factors of mother for the study of prevalence of birth injuries and associated factors.**

| Variables | Category | Frequency | Percentages (%) |
|---|---|---|---|
| **Fetal presentation** | Vertex presentation | 342 | 91.7 |
| | Breech presentation | 23 | 6.2 |
| | Face presentation | 5 | 1.3 |
| | Brow presentation | 3 | 0.8 |
| **Intrapartal fetal distress** | Yes | 88 | 23.6 |
| | No | 285 | 76.4 |
| **CPD** | Yes | 9 | 2.4 |
| | No | 364 | 97.6 |
| **Condition of labor** | Spontaneous | 254 | 68.1 |
| | Induced | 60 | 16.1 |
| | No labor (elective c/s) | 59 | 15.8 |
| **Duration of labor** | Normal | 195 | 52.3 |
| | Prolonged | 119 | 31.9 |
| | No labor | 59 | 15.8 |
| **Premature rupture of membrane** | Yes | 90 | 24.1 |
| | No | 283 | 75.9 |
| **Duration of rupture of membrane** | <18 hours | 317 | 85 |
| | $\geq$ 18 hours | 56 | 15 |
| **Color of amniotic fluid** | Clear | 292 | 78.3 |
| | Meconium stained | 81 | 21.7 |
| **Mode of delivery** | SVD | 119 | 31.9 |
| | Instrumental delivery | 37 | 9.9 |
| | C/S | 217 | 58.2 |
| **Cord prolapse** | Yes | 8 | 2.1 |
| | No | 365 | 97.9 |
| **Tight nuchal cord** | Yes | 13 | 3.5 |
| | No | 360 | 96.5 |
| **Qualifications of birth attendant** | Gynecologists/obstetricians | 54 | 14.5 |
| | Residents | 184 | 49.3 |
| | Midwifes | 135 | 36.2 |
| **Time of birth** | Day time birth | 230 | 61.7 |
| | Night time birth | 143 | 38.3 |

**Key**: CPD: Cephalopelvic Disproportion, C/S: Caesarian Section, SVD: Spontaneous Vaginal Delivery

## The associated factors of birth asphyxia

In multivariable logistic regression analysis, short height of the mothers, intrapartal fetal distress, cord prolapse, tight nuchal cord, birth attended by residents, male sex and low birth weight of the newborns were the most contributing factors of birth asphyxia (Table 6).

The occurrence of birth asphyxia was 10.7 times (AOR = 10.7, 95% CI: 3.59–32.4) higher to occur among neonates born from mothers with short height (<145 cm) in relative to neonates born from mothers with height >145 cm. Similarly, the odds of birth asphyxia among mothers who had intrapartal fetal distress were nearly five times (AOR = 4.74, 95% CI: 1.81–12.4) higher than their counterpart. Furthermore, newborns who had cord prolapse and nuchal cord during delivery were 7.7 times (AOR = 7.7, 95% CI: 1.45–34.0) and 9.2 times (AOR = 9.2, 95% CI: (4.9–35.3) more likely experienced birth asphyxia compared to those neonates born without cord prolapse and nuchal cord respectively.

Table 4. Early neonatal related factors of newborns delivered in public hospitals, Addis Ababa, Ethiopia, 2021 (n = 373).

| Variables | Category | Frequency | Percentages (%) |
|---|---|---|---|
| **Sex** | Male | 225 | 60.3 |
| | Female | 148 | 39.7 |
| **Gestational age** | <37 weeks (preterm) | 44 | 11.8 |
| | 37–42 weeks (term) | 288 | 77.2 |
| | >42 weeks (post term) | 41 | 11 |
| **Birth weight** | <2500 gram | 54 | 14.5 |
| | 2500–3999 gram | 285 | 76.4 |
| | ≥4000 gram | 34 | 9.1 |
| **Head circumference** | <33 cm | 21 | 5.6 |
| | 33–38 cm | 336 | 90.1 |
| | >38 cm | 16 | 4.3 |
| **Cry after birth** | Yes | 321 | 86.1 |
| | No | 52 | 13.9 |
| **APGAR score (1st minutes)** | 0–3 (low) | 17 | 4.6 |
| | 4–6 (moderate) | 62 | 16.6 |
| | 7–10 (normal) | 294 | 78.8 |
| **APGAR score (5th minutes)** | 0–3 | 9 | 2.4 |
| | 4–6 | 43 | 11.5 |
| | 7–10 | 321 | 86.1 |
| **Resuscitation after birth** | Yes | 52 | 13.9 |
| | No | 321 | 86.1 |

Labor attended by residents were 81% less likely (AOR = 0.19, 95% CI: 0.05–0.68) to encounter birth asphyxia among newborns compared to those labor attended by gynecologist/ obstetricians. Besides, the odds of experiencing birth asphyxia was nearly four times higher (AOR = 3.84, 95% CI: 1.30–11.3) among male newborns comparing to female newborns. In addition to this, low birth weight newborns were 5.28 more likely (AOR = 5.28, 95% CI: 1.58–17.6) to develop birth asphyxia relative to normal birth weight newborns (Table 6).

## The associated factors of birth trauma

To control the effect of confounding, multivariate analysis were done and factors independently associated with birth trauma were GDM, prolonged duration of labor, instrumental delivery and night time birth (Table 7).

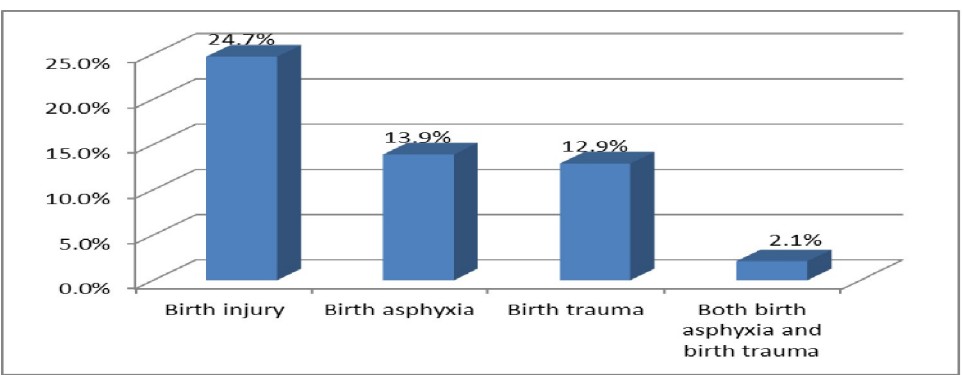

**Fig 1. Prevalence of birth injury among newborns delivered in public hospitals Addis Ababa, Ethiopia, 2021.**

**Table 5. Common types of birth trauma among newborns delivered in public hospitals, Addis Ababa, Ethiopia, 2021.**

| Types of birth trauma | Frequency (n) | Percentages (%) from newborn with birth trauma (n = 48) | Percentages (%) from study population (n = 373) |
|---|---|---|---|
| **Extra cranial trauma** | | | |
| Caput succedaneum | 9 | 18.8 | 2.41 |
| Cephalhaematoma | 10 | 20.8 | 2.68 |
| Subgalial hemorrhage | 20 | 41.7 | 5.36 |
| **Neurologic trauma** | | | |
| Erb's palsy | 5 | 10.4 | 1.3 |
| Facial palsy | 8 | 16.7 | 2.1 |
| **Soft tissue trauma** | | | |
| Facial and skin bruises | 5 | 10.4 | 1.3 |
| Skin laceration | 3 | 6.3 | 0.8 |
| Sub-conjuctival hemorrhage | 2 | 4.2 | 0.5 |

The odds of birth trauma were 5 times (AOR = 5.01, 95% CI: 1.38–18.31) higher among neonates born from mothers with gestational diabetic mellitus compared to those born from mothers who did not experience gestational diabetic mellitus. Regarding duration of labor, neonates born from mothers who had prolonged labor were 3.74 times (AOR = 3.74, 95% CI: 1.52–9.20) more likely to develop birth trauma when compared to those born from mother with normal duration of labor. Those neonates born via instrumental assisted were nearly 10.6 times (AOR = 10.6, 95% CI: 3.45–32.7) more susceptible to experience birth trauma than neonates delivered via caesarian section. Moreover, neonates delivered during the night time were nearly five times (AOR = 4.82, 95% CI: 1.84–12.6) more likelihood of acquiring birth trauma than neonates born during the day time (Table 7).

## Discussion

Birth injury is the primary cause of morbidity among live birth newborns in our study area. The prevalence of birth injury differs widely from place to place. In this study the burdens and associated factors of birth injury among live birth newborns at Addis Ababa Public Hospital are reported. Short height of the mothers, intrapartal fetal distress, cord prolapse, tight nuchal cord, birth attended by residents, male sex and low birth weight where found to be a significant predictors of birth asphyxia. Whereas, birth trauma was significantly associated with gestational diabetic mellitus, prolonged duration of labor, instrumental delivery and night time birth.

### Prevalence of birth injury

The overall prevalence of birth injury among live birth newborns was 24.7% with 95% CI (20.1–29.0). It was higher than studies conducted in Indian, Iran, Nigeria and Jimma (11.76%, 2.2%, 5.7%, 15.4% respectively) [10, 12, 13, 18]. This variation might be due to difference in sample size and study area (this study conducted in referral hospitals where more complicated cases and referred from different setting that could increase the prevalence of birth injury in the study area).

In this study the prevalence of birth asphyxia was 13.9% with 95% CI (10.5–17.7). This finding was higher compared to studies conducted in Jimma 8.1% [18], Dire Dawa 2.5% [20] and South Indian 5.29% [13]. However, it was lower than the studies conducted in Jimma zone public hospitals 32.9% [19], Debre Tabor 28.35% [22], North East Amhara 22.6% [21] and Hossana 15.1% [8]. Similarly, this finding also lower as compared to the studies conducted in Iran 16.8% [10] and Nigeria 39.3% [12]. The possible reason might be difference in sample

**Table 6. Bivariate and multivariable logistic regression analysis for the associated factors of birth asphyxia.**

| Variables | Category | Birth asphyxia | | COR(95% CI) | AOR (95% CI) |
|---|---|---|---|---|---|
| | | Yes(n = 52) | No(n = 321) | | |
| **Age groups of mothers** | 15–19 | 4(7.7%) | 18(5.6%) | 1 | 1 |
| | 20–24 | 14(26.9%) | 74(23.1%) | 0.85(0.25–0.28) | 0.71(0.11–4.46) |
| | 25–29 | 19(36.5) | 122(38%) | 0.70(0.21–2.29) | 0.67(0.10–4.30) |
| | 30–34 | 4(7.7%) | 69(21.5%) | **0.26(0.05–1.14)**[*] | 0.59(0.06–5.48) |
| | ≥35 | 11(21.2%) | 38(11.8%) | 1.30(0.36–4.65) | 3.85(0.44–33.0) |
| **Educational status of mothers** | No formal education | 9(17.3%) | 42(13.1%) | **3.25(1.02–10.3)**[*] | 1.09(0.22–5.41) |
| | Primary | 21(40.4%) | 112(34.9%) | **2.85(1.03–7.88)**[*] | 1.38(0.37–5.03) |
| | Secondary | 17(32.7%) | 91(28.3%) | **2.84(1.00–8.05)**[*] | 1.33(0.32–5.51) |
| | Above secondary | 5(9.6%) | 76(23.7%) | 1 | 1 |
| **BMI (Kg/m$^2$)** | <18.5 | 4(7.7%) | 26(8.1) | 1 | 1 |
| | 18.5–24.9 | 26(50%) | 238(74.1%) | 0.71(0.23–2.19) | 1.06(0.18–6.06) |
| | 25–29.9 | 19(36.5%) | 49(15.3%) | **2.52(0.77–8.18)**[*] | 2.08(0.31–13.5) |
| | ≥30 | 3(5.8%) | 8(2.5%) | 2.43(0.44–13.2) | 3.06(0.29–32.4) |
| **Height of the mothers** | <145 cm | 22(42.3%) | 29(9%) | **7.38(3.78–14.4)**[*] | **10.7(3.59–32.4)**[**] |
| | ≥145 cm | 30(57.7%) | 292(91%) | 1 | 1 |
| **Parity** | Primipara | 31(59.6%) | 155(48.3%) | **1.58(0.87–2.86)**[*] | 2.04(0.72–5.77) |
| | Multipara | 21(40.4%) | 166(51.7%) | 1 | 1 |
| **GDM** | Yes | 10(19.2%) | 30(9.3%) | **2.31(1.05–5.06)**[*] | 2.24(0.52–9.67) |
| | No | 42(80.8%) | 291(90.7%) | 1 | 1 |
| **Types of pregnancy** | Single | 50(96.2%) | 291(90.7%) | **2.57(0.59–11.1)**[*] | 4.48(0.49–40.7) |
| | Twine | 2(3.8%) | 30(9.3%) | 1 | 1 |
| **Abruptio placenta** | Yes | 4(7.7%) | 4(1.2%) | **6.6(1.59–27.2)**[*] | 5.30(0.52–54.0) |
| | No | 48(92.3%) | 317(98.8%) | 1 | 1 |
| **Intrapartal fetal distress** | Yes | 26(50%) | 62(19.3%) | **4.17(2.26–7.68)**[*] | **4.74(1.81–12.4)**[**] |
| | No | 26(50%) | 259(80.7%) | 1 | 1 |
| **CPD** | Yes | 4(7.7%) | 5(1.6%) | **5.26(1.36–20.3)**[*] | 5.08(0.85–30.3) |
| | No | 48(92.3%) | 316(98.4%) | 1 | 1 |
| **Condition of labor** | Spontaneous | 42(80.8%) | 212(66%) | **3.78(1.13–12.6)**[*] | 6.73(0.88–51.2) |
| | Induced | 7(13.5%) | 53(16.5%) | **2.5(0.61–10.2)**[*] | 2.88(0.30–27.4) |
| | No labor | 3(5.8%) | 56(17.4%) | 1 | 1 |
| **Duration of labor** | Normal | 24(49%) | 171(64.5%) | 1 | 1 |
| | Prolonged | 25(51%) | 94(35.5%) | **1.89(1.02–3.5)**[*] | 1.80(0.70–4.62) |
| | No labor | 3(5.8%) | 56(17.4%) | **0.38(0.11–1.31)**[*] | 0.54(0.35–2.42) |
| **Duration of rupture of membranes** | <18 hours | 41(78.8%) | 276(86%) | 1 | 1 |
| | ≥18 hours | 11(21.2%) | 45(14%) | **1.64(0.78–3.43)**[*] | 1.05(0.33–3.32) |
| **Color of amniotic fluids** | Clear | 33(63.5%) | 259(80.7%) | 1 | 1 |
| | Meconium stained | 19(36.5) | 62(19.3%) | **2.4(1.28–4.51)**[*] | 1.95(0.72–5.27) |
| **Cord prolapse** | Yes | 4(7.7%) | 4(1.2%) | **6.6(1.59–27.2)**[*] | **7.7 (1.45–34.0)**[**] |
| | No | 48(92.3%) | 317(98.8) | 1 | 1 |
| **Tight nuchal cord** | Yes | 7(13.5%) | 6(1.9%) | **8.16(2.62–25.3)**[*] | **9.2 (4.9–35.3)**[**] |
| | No | 45(86.5%) | 315(98.1%) | 1 | **1** |
| **Qualification of birth attendant** | Gynecologist | 14(26.9%) | 40(12.5%) | 1 | 1 |
| | Residents | 25(48.1%) | 159(49.5%) | **0.44(0.21–0.94)**[*] | **0.19(0.05–0.68)**[**] |
| | Midwifes | 13(25%) | 122(38%) | **0.3(0.13–0.7)**[*] | 0.62(0.15–2.56) |
| **Time of birth** | Day time | 20(38.5%) | 210(65.4%) | 1 | 1 |
| | Night time | 32(61.5%) | 111(34.6%) | **3.02(1.65–5.53)**[*] | 1.81(0.73–4.51) |

(*Continued*)

**Table 6.** (Continued)

| Variables | Category | Birth asphyxia | | COR(95% CI) | AOR (95% CI) |
|---|---|---|---|---|---|
| | | Yes(n = 52) | No(n = 321) | | |
| **Sex** | Male | 39(75%) | 186(57.9%) | **2.17(1.11–4.23)**\* | **3.84(1.30–11.3)**\*\* |
| | Female | 13(25%) | 135(42.1%) | 1 | 1 |
| **Birth weight** | <2500 g | 12(23.1%) | 42(13.1%) | **2.10(1.01–4.39)**\* | **5.28(1.58–17.6)**\*\* |
| | 2500–3999 g | 34(65.4%) | 251(78.2%) | 1 | 1 |
| | ≥4000g | 6(11.5%) | 28(8.7%) | 1.58(0.61–4.09) | 0.29(0.04–1.75) |

Hosmer and Lemeshow test, P-value = 0.758.

\*statistically significant by COR at P-value ≤0.25.

\*\* Statistically significant by AOR at P-value<0.05.

**Key**: BMI: Body Mass Index, CPD: Cephalopelvic Disproportion, GDM: Gestational Diabetes Mellitus, COR: Crude Odds Ratio, AOR: Adjusted Odds Ratio

size, using different definition of birth asphyxia (some studies used 1st minutes APGAR score, but this study used 5th minutes APGAR score to define birth asphyxia), variation of the study area and variation in distribution of skilled birth attendant in different setting.

The current study showed that the prevalence of birth trauma was 12.9% with 95% CI (9.7–16.4). This finding was higher as compared to the studies done in USA 2.9% [15], Pakistan 4.11% [24], India 1.54% [16] and Jimma 8.1% [18], However, this result was lower than studies conducted in Nigeria 67.2% [25]. This might be due to difference in study design, sample size, study population and variation in diagnosis of birth trauma, i.e. this study used birth trauma that was diagnosed only by physical examination but other studies included birth trauma diagnosed by both physical examination and radiological.

The most common birth trauma seen in the current study was extra cranial trauma 39 (81.2%), neurological trauma 13 (27%) and soft tissue trauma10 (21%). Subgalial hemorrhage 41.7% and cephalhaematoma 20.8% were the most common birth trauma. This finding was higher than studies done in Jimma and Nigeria, they were found that the most prevailing birth trauma was subgalial hemorrhage which accounts 20% and 13.1% respectively. The possible reason might be in the current study, instrumental delivery is significantly associated with birth trauma but not in study conducted in Jimma [18]. In addition to this, there was low rate of instrumental assisted delivery due to fear of cultural belief, so most women prefer to deliver by spontaneous vaginal delivery in study conducted in Nigeria [12].

Cephalhaematoma was the second common types of birth trauma diagnosed in around 20.8% of the newborns, it was lower when compared to studies done in Iran [10] and India [26], they were found that the most common type of birth trauma was cephalhaematoma accounts 57.2% and 38.7% respectively. However this finding was higher than study done in Nigeria 16.4% [12] and Pakistan 2.14% [24]. This might be due to differs in the skill of birth attendant and frequency of instrumental delivery.

In this study, facial palsy was the most prevailing among neurological trauma. This finding was supported by studies carried out in Iran [10], Indian [16], Bombay Hospital [26] and Nigeria (Maiduguri) [12]. The possible reason may be the fact that facial palsy occur during difficult delivery when forceps are applied and leads to paralysis of seventh cranial nerve.

## The associated factors of birth asphyxia

Factors independently associated with birth asphyxia were short height of the mothers, intrapartal fetal distress, cord prolapse, tight nuchal cord, birth attended by residents, male sex of the newborns and low birth weight of the newborns.

**Table 7. Bivariate and multivariable logistic regression analysis for the associated factors of birth trauma.**

| Variables | Category | Birth trauma | | COR (95% CI) | AOR (95% CI) |
|---|---|---|---|---|---|
| | | Yes (n = 48) | No (n = 325) | | |
| BMI (Kg/m$^2$) | <18.5 | 3(6.3%) | 27(8.3%) | 1 | 1 |
| | 18.5–24.9 | 24(50%) | 240(73.8%) | 0.90(0.25–3.18) | 1.55(0.21–11.2) |
| | 25–29.9 | 18(37.5%) | 50(15.4%) | **3.24(0.87–11.9)**\* | 1.59(0.17–14.5) |
| | ≥30 | 3(6.3%) | 8(2.5%) | **3.37(0.56–20.0)**\* | 3.09(0.23–41.5) |
| Height of the mothers | <145 cm | 13(27.1%) | 38(11.7%) | **2.8(1.36–5.76)**\* | 1.73(0.54–5.55) |
| | ≥145 cm | 35(72.9%) | 287(88.3%) | 1 | 1 |
| Number of ANC follow up | 1–3 | 10(20.8%) | 45(13.8%) | 1 | 1 |
| | ≥4 | 36(75%) | 276(84.9%) | **0.58(0.27–1.26)**\* | 0.37(0.13–1.10) |
| GDM | Yes | 16(33.3%) | 24(7.4%) | **6.27(3.02–13.0)**\* | **5.01(1.38–18.1)**\*\* |
| | No | 32(66.7%) | 301(92.6%) | 1 | 1 |
| Fetal presentation | Vertex | 41(85.4%) | 301(92.6%) | **0.06(0.006–0.76)**\* | 0.04(0.002–1.08) |
| | Breech | 1(2.1%) | 22(6.8%) | **0.02(0.001–0.51)**\* | 0.11(0.002–5.55) |
| | Face | 4(8.3%) | 1(0.3%) | 2.00(0.07–51.5) | 3.36(0.05–21.7) |
| | Brow | 2(4.2%) | 1(0.3%) | 1 | 1 |
| Duration of labor | Normal | 19(36.6%) | 176(54.2%) | 1 | 1 |
| | Prolonged | 29(60.4%) | 90(27.7%) | **2.98(1.58–5.61)**\* | **3.74(1.52–9.20)**\*\* |
| | No labor | 0(0%) | 59(18.2%) | | |
| Mode of delivery | SVD | 13(27.1%) | 106(32.6%) | 1.44(0.65–3.08) | 1.15(0.39–3.32) |
| | Instrumental | 18(37.5%) | 19(5.8%) | **11.1(4.94–25.1)**\* | **10.6(3.45–32.7)**\*\* |
| | C/S | 17(35.4%) | 200(61.5%) | 1 | 1 |
| Time of birth | Day time | 13(27.1%) | 217(66.8%) | 1 | 1 |
| | Night time | 35(72.9%) | 108(33.2%) | **5.41(2.74–10.6)**\* | **4.82(1.84–12.6)**\*\* |
| Sex | Male | 34(70.8%) | 191(58.8%) | **1.7(0.88–3.29)**\* | 0.99(0.39–2.51) |
| | Female | 14(29.2%) | 134(41.2%) | 1 | 1 |
| Birth weight | <2500 g | 4(8.5%) | 50(15.4%) | 0.70(0.23–2.09) | 0.36(0.06–2.21) |
| | 2500–3999 g | 29(60.4%) | 256(78.8%) | 1 | 1 |
| | ≥4000 g | 15(31.3%) | 19(5.8%) | **6.96(3.20–15.1)**\* | 1.70(0.41–7.00) |
| Head circumference | <33 cm | 3(6.3%) | 18(5.5%) | 1 | 1 |
| | 33–38 cm | 36(75%) | 300(92.3%) | 0.72(0.20–2.56) | 0.12(0.01–1.09) |
| | >38 cm | 9(18.8%) | 7(2.2%) | **7.71(1.60–37.1)**\* | 1.25(0.09–17.1) |

Hosmer and Lemeshow test, P-value = 0.85.

\* = Statistically significant by COR at P-value ≤0.25.

\*\* = Statistically significant by AOR at P-value<0.05.

**Key:** ANC: Antenatal Care, BMI: Body Mass Index, C/S: Caesarian Section, GDM: Gestational Diabetes Mellitus, COR: Crude Odds Ratio, AOR: Adjusted Odds Ratio

The occurrence of birth asphyxia was 10.7 times (AOR = 10.7, 95% CI: 3.59–32.4) higher among neonates born from mothers with short height (<145 cm) in relative to neonates born from mothers with height >145 cm. This finding was supported by studies conducted in Swedish [27], Uganda [28] and Ethiopia [29]. This could be due to the fact that those mothers who had short height may have short stature that impair the progress of descent of the fetal head and leads to prolong the duration of labor. This predisposes the newborn for birth asphyxia.

Our study also identified that intrapartal fetal distress was significantly associated with birth asphyxia. The odds of birth asphyxia among mothers who had intrapartal fetal distress were nearly five times (AOR = 4.74, 95% CI: 1.81–12.4) higher as compared to those mothers

without history of intrapartal fetal distress. This finding was almost similar to the previous studies conducted in Gonder [30] and Addis Ababa [24]. The likely reason is either fetal tachy-cardia or fetal bradycardia is the main cause for fetal-placental oxygen deprivation that exposes the newborn for birth asphyxia. Usually it's an indication for emergency cesarean section. But this finding is lower than the study conducted in Jimma, Ethiopia neonates with intrapartal fetal distress had 6.4 times more likely to develop birth asphyxia when compare to neonates without intrapartal fetal distress [18]. This difference may be due to variation in study setting and quality of the obstetric care.

The occurrence of birth asphyxia was also independently associated with cord prolapse and tight nuchal cord. Newborns who had cord prolapse during delivery were 7.7 times (AOR = 7.7, 95% CI: 1.45–34.0) and tight nuchal cord during delivery were 9.2 times (AOR = 9.2, 95% CI: 4.9–35.3) more likely experienced birth asphyxia compared to their counterpart. This finding was supported with the previous studies conducted in USA [31], Hossana [8] and Jimma [19]. This could be due to the fact that compression of the cord may impair blood flow to the fetus and compromise the fetal oxygenation; as a result the chance of occurrence of birth asphyxia will be more likely.

Labor attended by residents were 81% less likely (AOR = 0.19, 95% CI: 0.05–0.68) to encounter birth asphyxia among newborns compared to those labor attended by gynecologist/ obstetricians. This might be due to since the study was conducted in teaching hospitals; most labor was attended by residents, but labor attended by gynecologists/obstetricians was critical cases/ consulted case that was unable to handle by residents. In addition to the above reason, there may be variation in skill of neonatal resuscitation b/n resident and gynacologist, that determine the newborns outcome [32]. This finding was inconsistent with study conducted in Debre Tabor, Ethiopia neonates delivered by Midwives 56.2% developed birth asphyxia [24]. The difference may be due to variation in study setting and distribution of skilled birth attendant i.e. trained in neonatal resuscitation.

The odds of experiencing birth asphyxia was nearly four times higher (AOR = 3.84, 95% CI: 1.30–11.3) among male newborns comparing to female newborns. This finding was supported by study conducted in Washington, American [33] and Ayder Hospital, Ethiopia [34]. This might be due to biological difference makes male more at risk for birth asphyxia and it needs further investigation. In addition to this, low birth weight newborns were 5.28 more likely (AOR = 5.28, 95% CI: 1.58–17.6) to develop birth asphyxia relative to those who had normal birth weight. It was in agreement with study conducted in Addis Ababa [35], Gonder [30] and Jimma [19]. This might be clarified by the fact that most low birth weight neonates delivered during preterm gestation that might have immature lung and unable to pass the transition period without difficulty of breathing.

## The associated factors of birth trauma

The other dependent variable is birth trauma and the associated factors were found to be GDM, prolonged duration of labor, instrumental delivery and night time birth. The odds of birth trauma were 5 times (AOR = 5.01, 95% CI: 1.38–18.1) higher among neonates born from mothers with gestational diabetic mellitus compared to those born from mothers who did not experience gestational diabetic mellitus. This finding was consistent with the studies conducted in Nigeria [25] and Turkey [36]. This might be due to the truth that, one of the complications of infant of diabetic mothers is macrosomia, and this will predispose the newborn for mechanical birth trauma that is why it's the main reason for emergency C/s.

Neonates born from mothers who had prolonged labor were 3.74 times (AOR = 3.74, 95% CI: 1.52–9.20) more likely to develop birth trauma when compared to those born from mother

with normal duration of labor. This finding was supported by studies done in Nigeria [25], Indian [16] and Bombay hospital [26]. This is due to the fact that when there is prolonged labor, the women may experience tiredness and unable to progress the labor. Therefore, to prevent fetal distress, the birth attendant may apply forceps or vacuum to assist the labor. All these difficulty may leads to birth trauma.

Another contributing factor significantly associated with birth trauma was instrumental delivery. Those neonates born via instrumental assisted were 10.6 times (AOR = 10.6, 95% CI: 3.45–32.7) more susceptible to experience birth trauma than neonates delivered via cesarean section. This finding was in agreement with studies conducted in Bombay Hospital [26], Indian [13] and Nigeria [12]. The likely reason was due to the fact that, application of forceps and vacuum on the fetal head may expose to extra cranial hemorrhage, intra cranial hemorrhage and soft tissue abrasion/laceration. All these complication may leads to birth trauma. But, this finding was higher than study done in Pakistan [24], neonates delivered by instrument assisted were 2.14 times (AOR = 2.14) more likely to develop birth trauma than neonates delivered via cesarean section. This difference might be due to variation in study setting and skill of birth attendant.

Night time delivery was another contributing factor for birth trauma. Neonates delivered during the night time were nearly five times (AOR = 4.82, 95% CI: 1.84–12.6) more likelihood of acquiring birth trauma than neonates born during the day time. This finding was supported by study conducted Indian [16]. This is possibly justified by the number of birth attendant assigned during duty hours were few that makes them unable to accomplish the overburden during night time, expert in the field/gynecologist may not arrived on time for consulted cases and it might be large proportion of referred cases during night time.

## Conclusion and recommendation

The overall prevalence of birth injury in this study was 24.7%, which is still higher than the previous studies conducted in developing countries. Each birth asphyxia and birth trauma constitutes 13.9% and 12.9% respectively. Birth asphyxia was independently associated with short height of the mothers, intrapartal fetal distress, cord prolapse, tight nuchal cord, birth attended by residents, male sex of the newborns and low birth weight of the newborns. In addition to this, birth trauma was independently associated with GDM, prolonged duration of labor, instrumental delivery and night time birth. However, the finding of this study could only be generalized to this cohort womens–newborns in the study setting. The medical service provided to the mothers and newborns during delivery is important to reduce the overall prevalence of birth injury and its burden.

Therefore, most of the above contributing factors are preventable and strong effort must be done to improve prenatal care and the delivery service which are vital to reduce the occurrence of birth injury and its complications.

## Supporting information

**S1 Data. Raw data of the study.**
(SAV)

**S1 File. Data collection tools used to assess prevalence of birth injuries and associated factors among newborns delivered in public hospitals Addis Ababa, Ethiopia, 2021.** Crossectional study.
(DOCX)

## Acknowledgments

We would like to acknowledge department of Nursing, School of Nursing and Midwifery, Addis Ababa University and St.Peter Specialized Hospital. Our deepest appreciation and thanks also extend to Mr. Bereket G/Michael and Dr. Asrat Demtse for their unreserved guidance and support throughout this work. My gratitude will also extend to data collectors, study participants and supervisors for their supports and commitments to gather data.

## Author Contributions

**Conceptualization:** Esubalew Amsalu Tibebu, Kalkidan Wondwossen Desta, Feven Mulugeta Ashagre, Asegedech Asmamaw Jemberu.

**Data curation:** Esubalew Amsalu Tibebu, Kalkidan Wondwossen Desta, Feven Mulugeta Ashagre, Asegedech Asmamaw Jemberu.

**Formal analysis:** Esubalew Amsalu Tibebu.

**Methodology:** Esubalew Amsalu Tibebu, Kalkidan Wondwossen Desta, Feven Mulugeta Ashagre, Asegedech Asmamaw Jemberu.

**Software:** Esubalew Amsalu Tibebu.

**Supervision:** Kalkidan Wondwossen Desta, Feven Mulugeta Ashagre, Asegedech Asmamaw Jemberu.

**Writing – original draft:** Esubalew Amsalu Tibebu, Asegedech Asmamaw Jemberu.

**Writing – review & editing:** Esubalew Amsalu Tibebu, Kalkidan Wondwossen Desta, Feven Mulugeta Ashagre, Asegedech Asmamaw Jemberu.

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
