## [Decision Letter · Decision Letter 0]

14 Dec 2022

PONE-D-22-05977Prevalence of birth injuries and associated factors among newborns delivered in public hospitals Addis Ababa, Ethiopia, 2021. Crossectional StudyPLOS ONE

Dear Dr. Esubalew Amsalu Tibebu,

Thank you for submitting your manuscript to PLOS ONE. After careful consideration, we feel that it has merit but does not fully meet PLOS ONE’s publication criteria as it currently stands. Therefore, we invite you to submit a revised version of the manuscript that addresses the points raised during the review process.

We look forward to receiving your revised manuscript.

Kind regards,

Sanjoy Kumer Dey, M.D

Academic Editor

PLOS ONE

Journal Requirements:

Additional Editor Comments:

Dear Esubalew Amsalu Tibebu

Please go through reviewer's comment and address accordingly.

Reviewers' comments:

Reviewer's Responses to Questions

**Comments to the Author**

1. Is the manuscript technically sound, and do the data support the conclusions?

Reviewer #1: Yes

Reviewer #2: Partly

2. Has the statistical analysis been performed appropriately and rigorously? 

Reviewer #1: Yes

Reviewer #2: No

3. Have the authors made all data underlying the findings in their manuscript fully available?

Reviewer #1: No

Reviewer #2: No

4. Is the manuscript presented in an intelligible fashion and written in standard English?

Reviewer #1: Yes

Reviewer #2: Yes

5. Review Comments to the Author

Reviewer #1: The topic is interesting and has public health importance but there are issues needs to address

Your out come variable is not clear so you should specify whether Birth injuries or Birth trauma better to focus on over all Birth injuries

One of your variables was pre pregnancy BMI, do you think the participant remember pre pregnancy BMI? How do you asses quality of birth attendant?

Ethical clearance section needs further elaboration especially how did you maintain confidentiality, privacy and linking cases to the specific services and so on.

How do you assess the data of intra-partum and early neonatal variables using checklist?

What are others in HIV status result other than positive and negative result?

You mentioned that night time delivery is associated with birth trauma and the justification seems not scientific i.e. cases are referred during night time and duty issue what if the pregnant referred at the day time and deliver during night

The medical service provided to the mothers and newborns during delivery is important to reduce the overall prevalence of birth injury and its burden. Is it recommendation or discussion?

Your recommendation and conclusion should be presented from your pertinent findings and address potential associated factors.

Reviewer #2: Dear Respectable Editor, I am grateful to give me the opportunity to review the manuscript entitled “Prevalence of birth injuries and associated factors among newborns delivered in public hospitals Addis Ababa, Ethiopia, 2021. Crossectional Study” by Tibebu et al.

The manuscript aimed to report the prevalence and possible factors associated with birth injuries in a public hospitals in Ethiopia. The paper had the merit to address a topic of fundamental importance in Obstetric care in low-income country and to identify possible risk factors in a real-world setting. Nonetheless some issue can be addressed to improve the quality:

1) I suggest to shorten the introduction, to avoid repetition and to move to discussion the passage from line 80 to 94.

2) In the method section I advice to not provide as a separated paragraph the simple size and the procedure removing mathematical formula and summarizing in a single sentence (i.e: the single population proportion formula was used to determine the sample size with the following assumptions […] taken in account the study […] with CI 95%).

3) The main problem in the method section and subsequently in the reported results was related to the variables considered and their definition. In the statistical analysis the authors used the total length of the labour but it would be desirable to report first and second stage separately. In fact, labor and birth is a dynamic process and each stage of labor is characterized by different aspects, risks, obstetrical management and impact on fetal and maternal wellbeing. The total duration of the labor does not represent an adequate parameter and statistical analysis loses of significance.

4) Moreover, a definition of intrapartum fetal distress is not adequately provided: “When the fetal heart rate is either <100 or >180 beat/minutes or if there is non reassurance fetal heart rate pattern” but instant measurement of fetal heart rate does not provide a definition of fetal distress and proper international guidelines need to be considered to define non-reassurance fetal heart.

5) The threshold of 1 hour for the premature rupture of membrane is totally arbitrary, add references if available

6) Apgar score is a subjective evaluation especially in non adequate trained health workers, I suggest to specify who made the evaluation (resident, midwife, gynecologist?).

7) In the results I suggest to avoid description of sociodemographic features and to refer to Table 1

8) Provide the list of abbreviations under each table to a better comprehension

9) In my opinion was essential to report the number of elective, urgent and emergency cesarean section because they were burdened of different level of birth injury risk. A subgroup analysis is desirable to understand the association with birth injury in the univariate and multivariate analysis.

10) I advice to provide a unique paragraph for the Discussion

11) The Discussion lacks a decent literature review, and it is unclear what the study adds to what is already known on the subject.

12) I suggest to provide a more deep analysis of the literature evidence and to analyze, moreover, the possible impact of healthcare personnel training ( See this paper: 10.11604/pamj.2022.42.169.32816 ).

13) In relation to the findings on the role of maternal height, fetal head circumference and pelvic-fetal disproportion consider this paper for the discussion 10.1371/journal.pone.0275400.

6. PLOS authors have the option to publish the peer review history of their article (what does this mean?). If published, this will include your full peer review and any attached files.

Reviewer #1: No

Reviewer #2: **Yes: **Filippo Alberto Ferrari

---

## [Author Response · Author response to Decision Letter 0]

12 Jan 2023

Manuscript PONE-D-22-05977

 Rebuttal Letter

We really thank the academic editor and two reviewers for their thoughtful inputs and valuable comments on our manuscript. The feedback provided by them has been help full to improve this manuscript and we are grateful for their input. Please kindly find below our response to point by point raised by the academic editor and reviewers.

We hope that we clearly addressed all of them, and that the manuscript will be now suited for publication.

Sincerely,

On behalf of all the four authors,

Esubalew Amsalu Tibebu.

Academic editor

Journal requirement

1. Plos one templates

We have checked the Plos one templates and made the adjustment to meet the journal requirements.

2. Copyedit your manuscript for language usage, spelling, and grammar.

We have tried to check for any error in language usage, spelling and grammar and make adjustment 

1. Is the manuscript technically sound, and do the data support the conclusions?

Reviewer #1: Yes

Reviewer #2: Partly

2. Has the statistical analysis been performed appropriately and rigorously?

Reviewer #1: Yes

Reviewer #2: No

We thank the reviewer for this assessment, during statistical analysis we used bivariate logistic regression analysis to check the association between each independent variable with dependent variable and were tried to do the statistical analysis appropriately. Multivariable logistic regression model analysis also used for those variables with p-value ≤ 0.25 in order to control the confounding factors. To check the correlation between independent variables, multi-colinearity (colinearity diagnostic taste) was done by using the value of variance inflation factors and tolerance. Hosmer and Lemeshow goodness of fit test and omnibus tests of model coefficients were done to test the fitness of the logistic regression in the final model.

3. Have the authors made all data underlying the findings in their manuscript fully available?

Reviewer #1: No

Reviewer #2: No

We thank the reviewer for this assessment and we made amendments as follows

 All relevant data are within the manuscript and its Supporting information files. 

4. Is the manuscript presented in an intelligible fashion and written in standard English?

Reviewer #1: Yes

Reviewer #2: Yes

Review Comments to the Author

Reviewer #1

Comment 1

Your outcome variable is not clear so you should specify whether Birth injuries or Birth trauma better to focus on over all Birth injuries.

Response 1: Thank you for your comment and as we mentioned in operational definition, birth injuries is defined as injury to newborns that occur during labor and delivery who has diagnosis either of birth trauma, birth asphyxia or both. 

We defined birth trauma as any physical injury to newborns during the entire birth process. 

Birth asphyxia is failure to initiate, sustain breathing and not crying at birth and diagnosed based on Apgar score <7 at 5th minutes. To assess factors independently associated with birth asphyxia and birth trauma, it’s better to make our outcome variables are two i.e. birth asphyxia and birth trauma.

Comment 2

One of your variables was pre pregnancy BMI, do you think the participant remember pre pregnancy BMI? 

Response 2: We are grateful to respond to your constructive comments and here are the responses. 

The variable pre-pregnancy BMI was obtained by measuring maternal height and taking maternal pre-pregnancy weight (Kg/m²). The maternal pre-pregnancy weight was taken in two ways. The first one was by asking maternal pre-pregnancy weight if she remembers and the second way was taken from chart review during first trimester weight record (<12 wks) if she didn’t remember her weight. 

Since our Exclusion criteria were those who have incomplete documentation (has no appropriate data that measure both maternal and early neonatal parameter). So that, if the mother didn’t remember her Prepregnancy weight and didn’t have first trimester antenatal follow up weight record, she had excluded from the study participant.

Comment 3

How do you asses quality of birth attendant?

Response 3: Thank you for your observation. It was not our objective to assess the quality of birth attendant; rather we were concerned on qualification of birth attendants. That is assessed by checklist of asking whether the birth attendant is Midwife, general practitioner, residents or gynecologist.

Comment 4: Ethical clearance section needs further elaboration especially how did you maintain confidentiality, privacy and linking cases to the specific services and so on?

Response 4: We thank the reviewer for this question

Study participants were asked for their willingness to participate in the study after explaining the purpose of the study. Then written informed consent was obtained from each participant. The privacy and confidentiality of information was strictly maintained by not writing the name of study participants on data collection tool. 

Comment 5

How do you assess the data of intra-partum and early neonatal variables using checklist?

Response 5: We thank the reviewer for this question 

The checklist was consists of a total of 23 questions that used to assess data on intra-partum variables (such as, fetal presentation, duration of labor, prolonged rupture of membrane, premature rupture of membrane, cephalopelvic disproportion, intra-partal fetal distress, meconium stained amniotic fluids, mode of delivery, instrument use during delivery, cord prolapse, nuchal cord, time of birth and qualification of birth attendant), and early neonatal variables (such as, sex, birth weight, gestational age, APGAR score, need of resuscitation, head circumference) were taken from chart review of pregnant women who delivered during data collection period by using structured checklist.

Comment 6

What are others in HIV status result other than positive and negative result?

Response 6: Thank you again for your observation.

# On table 2, Medical and obstetrics characteristics of the mother, the word ‘Others’ refers not to HIV status, rather it refers to Anemia, congestive heart failure, thrombocytopenia, asthma and hydronephrosis.

Comment 7

You mentioned that night time delivery is associated with birth trauma and the justification seems not scientific i.e. cases are referred during night time and duty issue what if the pregnant referred at the day time and deliver during night? 

Response 7: This very important observation is highly appreciated

The justification may not be scientific and it needs further study. However, when we come to our hospital set up in Addis Ababa, Ethiopia, night time hospital environment is absolutely different from day time due to the following reason.

1. The number of staff available during the night time is almost half of the day time b/c of duty payment issue of the government.

2. Majority of the mother give birth in our setup is during the night time.

3. Most of the staffs are tired, fatigued and give less attention to the mother.

Comment 8

The medical service provided to the mothers and newborns during delivery is important to reduce the overall prevalence of birth injury and its burden. Is it recommendation or discussion?

Response 8: Thank you for this comment and we have moved this paragraph to recommendation part.

Reviewer # 2

Comment 1

I suggest shortening the introduction, to avoid repetition and to move to discussion the passage from line 80 to 94.

Response 1: We understand and agree with your comment and we thank the reviewer for suggestion. We made correction based on your comment.

Comment 2

In the method section i advice to not provide as a separated paragraph the simple size and the procedure removing mathematical formula and summarizing in a single sentence.

Response 2: We thank the reviewer for constructive comment and we edited the paragraph as below

The single population proportion formula was used to determine the sample size with the following assumptions: Where; n=Sample size, Z= 95 % confidence level (Z α/2 = 1.96), α = Level of significance 5% (α= 0.05) and d= Margin of error 5% (d = 0.05).

The prevalence of birth trauma was (P) = 8.1% taken from the previous study done in Jimma University Specialized Hospital, South Western Ethiopia, and sample size was 125. The prevalence of birth asphyxia was (P) = 32.9 % taken from the previous study conducted in Jimma zone public Hospitals, South West Ethiopia, after comparing with other studies done in Ethiopia , After considering 10% non-response rate, the total sample size was 373. Finally, from the calculated sample size for the first and second dependent variables, the largest sample size was 373

Comment 3 

The main problem in the method section and subsequently in the reported results was related to the variables considered and their definition. In the statistical analysis the authors used the total length of the labour but it would be desirable to report first and second stage separately. In fact, labor and birth is a dynamic process and each stage of labor is characterized by different aspects, risks, obstetrical management and impact on fetal and maternal wellbeing. The total duration of the labor does not represent an adequate parameter and statistical analysis loses of significance.

Response 3: These is very important comment, However, during data collection we have only taken the total duration of true labor, it would be better if we were take the data by categorizing first and second stage of labor. 

Comment 4 

Definition of intrapartum fetal distress

Response 4: Fetal distress, defined as progressive fetal hypoxia and/or academia secondary to inadequate fetal oxygenation, is a term that is used to indicate changes in fetal heart patterns, reduced fetal movement, fetal growth restriction, and presence of meconium stained fluid.

Comment 5 

The threshold of 1 hour for the premature rupture of membrane is totally arbitrary, add references if available

Response 5: We admit that and tried to correct it.

Premature rupture of memberane is rupture of membrane of the amniotic sac and chorion occurred before onset of true labor.

Comment 6

Apgar score is a subjective evaluation especially in non-adequate trained health workers, I suggest to specify who made the evaluation (resident, midwife, gynecologist?)

Response 6: This is very important comment and we need to appreciate to this

The APGAR score was evaluated by trained health worker like resident and gynacologist.

Comment 7 

In the results I suggest to avoid description of socio-demographic features and to refer to Table 1.

Response 7: This is also very important comment; therefor it is corrected based on your suggestion with in the manuscript.

Comment 8

Provide the list of abbreviations under each table to a better comprehension.

Response 8: Thank you very much for this observation; we have written the abbreviations under each table.

Comment 9: In my opinion was essential to report the number of elective, urgent and emergency cesarean section because they were burdened of different level of birth injury risk. A subgroup analysis is desirable to understand the association with birth injury in the univariate and multivariate analysis.

Response 9: We would like to appreciate your point of view and thank the reviewer for this comment. The number of either elective or emergency cesarean section was burdened for different level of birth injury risk. However, during analysis time bivariate logistic regression was done to check the association b/n mode of delivery (c/s) with birth asphyxia and we found that no association b/n mode of delivery with that of birth asphyxia (p-value >0.25). Due to this reason we could not able to do multivariate analysis for this variable.

Comment 10: I advise to provide a unique paragraph for the discussion.

Response 10: We thank and accept the reviewer for this comment. The following paragraph was added on the discussion part. 

In this study the burdens and associated factors of birth injury among live birth newborns at Addis Ababa Public Hospital are reported. Short height of the mothers, intrapartal fetal distress, cord prolapse, tight nuchal cord, birth attended by residents, male sex and low birth weight where found to be a significant predictors of birth asphyxia. Whereas, birth trauma was significantly associated with gestational diabetic mellitus, prolonged duration of labor, instrumental delivery and night time birth.

Comment 11: The Discussion lacks a decent literature review, and it is unclear what the study adds to what is already known on the subject.

Response 11: Thank you. We were tried to look all the available literature related to our research, know some literature review are incorporated.

Comment 12: I suggest to provide a more deep analysis of the literature evidence and to analyze, moreover, the possible impact of healthcare personnel training (See this paper: 10.11604/pamj.2022.42.169.32816).

Response 12: Thank you the reviewer for this suggestion. We have seen those above paper and they are majorly focused on the effect of healthcare personnel training on neonatal resuscitation, on the other hand in our study when we collect data we take only the qualification of birth attendant i.e. General practitioner, Gyni resident or midwifery.

Comment 13: In relation to the findings on the role of maternal height, fetal head circumference and pelvic-fetal disproportion consider this paper for the discussion 10.1371/journal.pone.0275400.

Response 13: Thank you reviewer for giving us this direction, we have seen and considered this paper in our discussion.

---

## [Editor Report · Decision Letter 1]

16 Jan 2023

Prevalence of birth injuries and associated factors among newborns delivered in public hospitals Addis Ababa, Ethiopia, 2021. Crossectional Study

PONE-D-22-05977R1

Dear Dr. Esubalew Amsalu Tibebu

We’re pleased to inform you that your manuscript has been judged scientifically suitable for publication and will be formally accepted for publication once it meets all outstanding technical requirements.

Kind regards,

Sanjoy Kumer Dey, M.D

Academic Editor

PLOS ONE
---

## [Editor Report · Acceptance letter]

20 Jan 2023

PONE-D-22-05977R1 

Prevalence of birth injuries and associated factors among newborns delivered in public hospitals Addis Ababa, Ethiopia, 2021. Crossectional Study 

Dear Dr. Tibebu:

I'm pleased to inform you that your manuscript has been deemed suitable for publication in PLOS ONE. Congratulations! Your manuscript is now with our production department. 

Kind regards, 

on behalf of

Dr. Sanjoy Kumer Dey 

Academic Editor

PLOS ONE